# Partners' experiences of their loved ones' trauma and PTSD: An ongoing journey of loss and gain

**Rosie Powling**[1¤]*, **Dora Brown**[1], **Sahra Tekin**[2], **Jo Billings**[2]

1 School of Psychology, University of Surrey, Guildford, United Kingdom, 2 Division of Psychiatry, University College London, London, United Kingdom

¤ Current address: Clinical Education Development And Research (CEDAR), Department of Psychology, University of Exeter, Exeter, United Kingdom
* r.powling@exeter.ac.uk

## Abstract

### Background

Traumatic life events can have a profound impact on the physical and psychological wellbeing of not only those who directly experience them, but others who are indirectly affected, such as victims' partners.

### Aims

This study aimed to explore the experiences and views of partners of individuals who have a history of trauma and diagnosis of posttraumatic stress disorder (PTSD).

### Methods

In-depth semi-structured interviews were conducted with six partners of people who had experienced trauma and were diagnosed with PTSD and awaiting or receiving treatment at a specialist Trauma Service. The data was analysed using Interpretative Phenomenological Analysis.

### Results

One overarching theme resulted from the data: partners experienced trauma and PTSD as an ongoing journey of loss and gain. This was supported by three superordinate themes: making sense of the trauma and ensuing consequences, shifting identities, and accessing and experiencing outside resources. Partners' journeys were characterised by striving and struggling to make sense of the trauma and its ensuing consequences, whilst grappling with the identities of themselves, their partners and relationships shifting over time. Participants navigated their journeys in the context of external resources and support from friends, family, colleagues and professionals.

**Data Availability Statement:** Data supporting the findings of this study cannot be made generally available due to the personal and sensitive content of participants' accounts, in line with the requirements stipulated by the University College

London and Royal Holloway Research Ethics Committee. The Participant Information Sheet, which was used as part of the process of gaining participants' informed consent, and where relevant partners' informed consent, stated that 'Only the researcher and two Supervisors of the study will have access to the information collected.' Therefore, if we were to share the data publicly, we would be breaching the informed consent given by participants and their partners, as well as our ethical considerations and the robust methodology upon which favourable ethical approval from the National Research Ethics Service, relevant NHS Trust Research and Development committee, and the University of Surrey Faculty of Arts and Human Sciences was based. We have included this statement in the section 'Availability of data and materials' in order to signpost the reader if required: "To enquire about ethics and the governance review process, contact the University Ethics Committee at the University of Surrey on ethics@surrey.ac.uk."

**Funding:** The author(s) received no specific funding for this work.

**Competing interests:** The authors have declared that no competing interests exist.

## Conclusions

The results of this study highlight the need for greater information and support for partners of people with PTSD.

## Introduction

Traumatic and life-threatening events can have a profound impact on the physical and psychological wellbeing of those who experience them. To date, however, most research has focused on the impact upon the individual directly experiencing the trauma and less attention has been paid to the impact that the trauma, and traumatised person, can have on the wider family system. Social support is a significant protective factor in the development and severity of post-traumatic stress disorder (PTSD) [1, 2]; however, limited research has investigated the impact of trauma and PTSD on family members providing social support.

To date, few studies have explored the impact of a person's trauma and PTSD on their partner and relationships. Quantitative research has focused on specific traumas and populations at high risk of exposure to trauma, mainly combat-related trauma in male military personnel. For example, partners of veterans with PTSD can experience a range of mental health concerns including secondary traumatic stress, general psychological distress and caregiver burden [3–5]. In veterans with more severe PTSD symptoms, their partners have increased distress, lower levels of functioning and poorer mental health [6]. Sleep problems, dissociation and sexual problems can predict lower relationship satisfaction [7]. Similarly, depressive symptoms and hostility of military service personnel have been linked with marital dissatisfaction [8]. The first meta-analysis on PTSD and partner outcomes found a link between PTSD and decreased relationship quality, as perceived by partners in both military and civilian samples [9]. Partners of firefighters exposed to major trauma can experience symptoms such as major depression, panic disorder, and generalised anxiety disorder [10]. More recently, 72.5% of doctors' spouses reported psychological distress during the COVID-19 pandemic [11].

Qualitative studies have explored partners' experiences, likewise mostly concerning spouses of workers in high-risk occupational groups. This body of research indicates deleterious effects of being the partner of an individual with PTSD, such as experiencing general psychological distress [12], anxiety, depression, and social dysfunction [13], secondary stress [14], and social isolation [15].

The qualitative literature highlights further nuances and some potentially protective factors and positive outcomes. For example, spouses of law enforcement officers reported that remembering good times together and supporting their partner through bad times increased communication, plus shared activities helped to develop resilience post-trauma [16]. Humour might be adaptive for couples to cope with trauma [17]. In a qualitative study of frontline healthcare workers' family members during the COVID-19 pandemic [18], spouses reported being proud of their loved ones and their work, but feeling that families' needs and spouses' careers were forgotten and subjugated, and several spouses reported feeling vicariously traumatised by hearing first-hand what their partner had been exposed to at work.

Varied experiences have been described in studies exploring heterogenous traumas, not limited to high-risk occupational groups. A history of trauma may both increase and decrease experiences of cohesion, communication, and understanding in partners' interactions [19]. Relationship functioning may be affected by a range of interpersonal mechanisms concerning roles, boundaries, intimacy and coping, with trauma both bringing couples closer together and

distancing them apart [20]. Both of these studies used a broad definition of trauma and did not require the partner to have PTSD, yet still the authors emphasise the detrimental systemic impact of trauma and appeal for its greater recognition by services [19].

The literature has highlighted that healthcare providers, government and wider society need to improve their understanding around PTSD and its impact on relationships and partners [21]. Partners can feel invisible and alone due to organisations and society not understanding the impact of PTSD on families, which is a major obstacle to accessing help: developing knowledge and understanding may increase partners' help-seeking, increase social connectedness, and decrease the sense of invisibility that has thus far contributed to their stress and potentially hindered coping [21].

The literature exploring partners' in-depth views and experiences of trauma and PTSD remains relatively sparse, and there is a significant gap in the literature exploring the perspectives of partners from outside of high-risk occupational settings.

The present study sought to address these gaps in the literature by asking: What are the experiences and views of people who have a partner with a history of trauma and diagnosis of PTSD? The study aimed to investigate meaning-making within the context of an intimate relationship.

## Method

### Ethical approval

This study and all of its procedures were approved by the UK National Research Ethics Service (Reference: 11/LO/0526), the University of Surrey Research Ethics Committee (Reference: 625-PSY-11) and the relevant National Health Service (NHS) Trust Research and Development Committee (Reference: PF482). All participants gave written informed consent to take part in the research.

### Participants

Participants were recruited through a specialist Trauma Service in the UK using a purposive sampling strategy. Individuals were eligible for the study if their partner was a person using the specialist Trauma Service (i.e. a service user) who was diagnosed with PTSD following trauma and awaiting, or in the process of receiving, psychological treatment for PTSD. The partners of individuals invited to participate had experienced a traumatic event during their relationship, remained in the relationship since and were cohabiting at the time of recruitment. Posttraumatic stress disorder following a type 1 trauma, that is a discrete, single-incident traumatic event [22], or multiple type 1 traumas, needed to be the primary difficulty for which treatment was being sought. However, comorbid diagnoses did not exclude individuals from the study. The study was interested in individuals' experiences irrespective of their partner's occupation, and sought to include a range of participants whose partners had experienced different types of trauma. Participants needed to be English-speaking adults (aged 18 years or over).

### Procedure

The research took place within a specialist NHS Trauma Service in the UK. A Service User and Carer panel at the University of Surrey was consulted regarding ethical considerations, all study materials and the interview schedule. These were developed in line with their feedback, which centred around the recruitment process, process of gaining informed consent from participants, and where relevant their partners, and managing the impact of the interviews on both participants and the researcher. In light of feedback from the panel, time was

incorporated for participants to reflect on their experience at the end of the interview and where appropriate to signpost participants to further support for their own wellbeing.

Potential participants were approached directly by the clinician involved in the care of their partner if they were already known to the clinician. For partners not in direct contact with the service, a double layer consent process was followed, whereby valid consent from both the service user and partner was obtained, and participants were invited to participate by their partner. Potential participants were provided with an information sheet about the research via the clinician (single layer consent) or partner (double layer consent), and invited to contact the researcher if they were interested in taking part in the study.

Written informed consent was obtained from service users for those partners not in contact with the service, and for all participants, prior to interview. Interviews were conducted in person by the first author (RP), who was a Trainee Clinical Psychologist at the time, with experience of working with service users experiencing PTSD primarily using a Cognitive Therapy for PTSD model [23], and a special interest in the systemic effects of trauma and PTSD.

A semi-structured interview schedule was developed, based on reading of the literature and the research team's clinical experiences. This was used to explore participants' experiences of their loved ones' trauma and PTSD by asking open-ended questions about their relationship and life, before and after the traumatic event(s) (see S1 File for interview schedule). Interviews lasted between one-and-a-half and three-and-a-half hours.

Throughout the research process, the researchers endeavoured to take several steps to maximise credibility, following the guidelines for qualitative research suggested by Yardley [24]. Sensitivity to context was facilitated by contextualising the results within previous research, considering themes that concorded, challenged and informed existing theory. Commitment and rigour were achieved by maintaining a high degree of attentiveness to individuals' stories and the broader, shared experiences of participants, attending peer-led Interpretative Phenomenological Analysis (IPA) and facilitator-led qualitative research groups (attended by RP), and utilising supervision. Transparency and coherence were facilitated through producing a coherent argument of the findings, including ambiguities, contradictions and exceptions. Verbatim extracts allow the reader to check interpretations. Throughout the process, the first author (RP) wrote a reflective diary in an attempt to enhance reflection and reflexivity, and bring her prior beliefs, assumptions, biases and concerns into consciousness, so these could be held in mind to ensure interpretations were grounded in the data [25].

## Analysis

Interviews were audio-recorded and transcribed verbatim by the first author (RP) with any identifying features of participants and their families removed. Participants, partners and family members were given pseudonyms which are used throughout the presentation of the results.

Data was analysed following the principles of IPA [25–27]. This method focuses on individuals' emotions, cognitions and meaning-making and is particularly well-suited to exploring how people make sense of significant life events [28]. Each transcript was read several times and coded by the first author. This coding included summaries, associations, commonalities, contradictions and preliminary interpretations. Potential themes that arose in the data were documented. These initial themes were key words that captured the essential quality of the text [26], striking a balance between the specific and abstract, grounded and conceptual [25]. Each transcript was analysed in turn in an iterative process, according to the principles of IPA. Themes were developed, checked against earlier transcripts, and adapted, in order to identify meaningful connections and find the most appropriate fit with participants' experiences both

individually and as a group. Other members of the research team (JB and DB) independently read portions of the transcripts and coding, and the development of themes was discussed iteratively as a research team.

## Results

Six partners of service users participated in the study. The mean age of participants was 51 years (range 38–67 years). All described themselves as white British. Four participants were working full-time; one was working part-time and one was semi-retired. Two couples had children. One couple were in a same-sex relationship; the others described themselves as in a heterosexual relationship. Participants had been in their relationship for a mean average of 20 years (range 3–28 years). Participants' partners had experienced a range of traumatic events; where partners had experienced multiple traumas, a primary trauma was identified, that is the trauma that participants perceived as having the biggest impact on their partner. The mean length of time since the primary trauma occurred was 8 years (range 2–20 years) and mean length of time since PTSD was diagnosed was 2 years (range 1–4 years). See Table 1 for further sociodemographic information of participants.

Whilst there was rich diversity amongst participants' stories, one major commonality was observed within the data. This final, overarching theme, related to participants' experiences of their loved ones' trauma and PTSD being "An ongoing journey of loss and gain". The main and supporting themes are described below (see Table 2) and illustrated with participant quotations.

### An ongoing journey of loss and gain

This overarching theme encapsulated all the participants' experiences. Participants perceived the trauma and PTSD as resulting in numerous losses and gains for themselves, their partners and their relationship over the course of time. These included losses and gains in understanding, perceived identity and support that shifted and evolved in various ways. Participants' experiences were interpreted as journeys with an unknown, unpredictable course that they traversed following the trauma. At times, participants felt they were venturing upon their journey

**Table 1. Participants' sociodemographic information.**

| Participant Pseudonym | Gender | Age | Marital Status | Partner's Traumatic Event (s)[1] | Length of Relationship (years) | Time Since Primary Trauma (years)[2] | Time Since PTSD Diagnosis (years) |
|---|---|---|---|---|---|---|---|
| Duncan | Male | Early fifties | Married | Physical injury | 28 | 3 | 2 |
| Kerry | Female | Late forties | Married | Physical injury (physical assault) | 25 | 13 | 1 |
| Eve | Female | Late forties | Married | Burns injury (physical health problem) | 23 | 7 | 1 |
| Roy | Male | Late sixties | Married | Sexual assault (physical injury) | 25 | 20 | 4/5 |
| Laura | Female | Late thirties | Cohabiting | Combat-related traumas | ~2.7 | ~2.4 | 1.5 |
| Sally | Female | Early fifties | Cohabiting | Experiences in intensive care | 16 | 5 | 1 |

[1] In cases of multiple traumatic events, the primary trauma was defined as the trauma that participants perceived as having the biggest impact on their partner. This is located outside of parentheses.

[2] In cases of multiple traumatic events, time reported relates to the trauma participants perceived as having the biggest impact on their partner.

**Table 2. Overarching, superordinate and sub-themes.**

| Overarching theme: | An ongoing journey of loss and gain | | |
|---|---|---|---|
| Superordinate themes: | 1. Making sense of the trauma and ensuing consequences | 2. Shifting identities | 3. Accessing and experiencing outside resources |
| Sub-themes: | • Striving and struggling to understand<br>• Entering a new perceived reality<br>• Looking backwards: reflecting and reminiscing<br>• Looking forwards: hope versus uncertainty | • Adapting to new responsibilities and roles<br>• Subordinated in the relationship<br>• The emotional journey | • Support from others<br>• Professional support |

alone; at others, they had the company of their partner and loved ones. To help explain the overarching theme, three superordinate themes were found to support it: 1. *making sense of the trauma and ensuing consequences*, 2. *shifting identities* and 3. *accessing and experiencing outside resources*.

**1. Making sense of the trauma and ensuing consequences.** Participants appeared both to strive and struggle to understand the traumatic event, PTSD and other ensuing consequences.

*Striving and struggling to understand*. All participants described aspects of their striving and struggling to understand the trauma and their partners' response to it. It was not just the PTSD that participants had to deal with but a wide range of consequences including depression, litigation, threats to their housing situation and financial strain.

For most participants, receiving the diagnosis of PTSD seemed to be a relief: a helpful experience that facilitated understanding. For some, it offered justification for their partners' difficulties. Diagnosis offered hope of progressing forward:

*Ah! It's got a name, so I can deal with that! But if it hasn't got a name, then you're stuffed aren't you? (Sally)*

Participants described how family, friends and the public failed to understand their partners' PTSD. It seemed to be the hidden nature of mental health difficulties that made understanding and empathising difficult for themselves and others, which both Laura and Kerry related to stigma. Some participants compared their partners' psychological injuries with a physical injury, intimating how much easier they thought life would be if the injury was physical:

*I thank my lucky stars that Matt's in one piece and alive, but to be honest if he was missing a leg, this would probably be a hell of a lot easier to deal with. . .If he was missing a leg, people can see that he's been injured in combat, but the fact that he can't walk down the street without being like a terrified little, you know, terrified child, people just don't have time for it, you know, they don't get it. (Laura)*

Information regarding psychological and physical trauma represented in the media served as points of reference for participants' understanding. For some, the struggle to understand PTSD seemed impeded rather than facilitated by the media, because they thought that PTSD occurred only in military contexts. Some participants had read multiple books on PTSD, enhancing their understanding on an intellectual level, but many reflected that they were still unable to understand fully and empathise with their partners.

*Entering a new perceived reality*. The trauma and PTSD appeared to signify major changes in participants' lives, including changes in their quality of life, physical health and relationships. Most participants described this reality as harder to live with or unwished for. There was

a sense of the PTSD being omnipotent, often in control of the relationship and causing conflict.

For some participants, there was a strong sense that the emergence of trauma and PTSD shattered their expectations surrounding their life course and put life on hold. Laura externalised the PTSD, comparing it to a "big black cloud" and later an "evil witch" who arrived just as she was starting to think her dreams would materialise:

*The PTSD has interfered in our life. It's like some big, nasty, evil witch or something that's come in and destroyed everything, all our hopes and dreams basically. Just when I finally thought, "This is it, the rest of my life, wahay, finally! We can look forward to a nice, long, happy rest of life together". It's just been trodden on and squashed and thrown out of the window.*

Most participants reported arguing more with their partners following the trauma. Some seemed to have largely overcome this, whilst others described conflict as an ongoing, significant element in the relationship. Participants also described the PTSD as having a detrimental effect on their social lives, whereby they socialised less frequently and plans were thwarted. In this way, participants had entered a social reality controlled by the PTSD:

*It stopped us socialising really . . .She wouldn't go out anywhere. . .I couldn't meet up with all the old friends . . .She would never go to a pub again . . .So our social life went downhill. (Roy)*

*Looking backwards*: *Reflecting and reminiscing.* Participants compared and contrasted life at present with life pre-trauma and post-trauma. All participants reported significant changes, but they varied hugely in terms of how they viewed these changes. A range of losses or threatened losses were highlighted, including the loss of their partner, relationship and career, and there was often a strong sense of sadness or mourning over these losses:

*I miss him desperately. It's like I'm grieving, it's like I'm in mourning, but he's still alive. It's difficult to grieve for somebody when you see them day in, day out. And it's hard to reconcile that in your head that I'm in mourning for a loss, but he's still there. (Laura)*

In looking backwards, all participants commented upon changes in the emotional connection with their partner. A negative impact upon intimacy and the sexual dimension of relationships was described. Some said that the PTSD had affected their partners' ability to communicate with them, resulting in participants feeling angry, upset, unsympathetic or more distant from their partner.

Whilst most participants described various difficulties the trauma had generated in their relationships, two described how their relationship had strengthened over time, despite these strains. In particular, Duncan perceived the trauma and PTSD as having primarily positive consequences and likened his relationship to a journey, which was reset "in the right direction" through increased quality time and communication with his wife:

*We try to go for a walk practically every day round the park. . .which actually has brought us closer together, just that time to walk together and perhaps sit on the bench for a while and talk things through. So that's actually been very, very positive.*

*Looking forwards*: *Hope versus uncertainty.* Participants contemplated their future lives and relationship with hope, as well as uncertainty, fear and at times hopelessness. Visions of the

future varied from envisaging the relationship becoming stronger and the future being an ideal time of their lives financially and socially, to hope for greater equality in the future relationship, to simply hoping for life to return to how it was previously. Sally wanted her future to involve doing "normal mundane things" again, such as seeing friends together, whilst Duncan appeared particularly hopeful about the future.

Participants' uncertainty about the future often concerned whether PTSD was too large an obstacle to conquer. Several participants seemed to experience a dialectic of holding hope, yet fearing "this is it". A few participants feared their partner might commit suicide; two had previously attempted suicide. The following extract demonstrates how Laura swung between feeling hopeful and fearful about whether she would regain her partner, whom she viewed as lost, or whether PTSD would win the battle:

> I can't help but feel [the PTSD] is a temporary thing and that it will be dealt with and he will be treated and that he will get over it and he'll get better. . .But equally, I can't help but worry that that's a bit optimistic and that he won't ever be back to his old self and that this is how it is.

For all participants, there was a sense of commitment to their partners and embarking upon the future together. However, a more complex narrative emerged in some interviews, with a strong sense of commitment struggling to coincide alongside uncertainty about whether or not the relationship could continue.

**2. Shifting identities.** Alongside the endeavour to make sense of everything, there was a strong sense of the identities and concomitant roles of participants and their partners evolving as a result of the trauma. This occurred as participants strived to cope with new positions in their relationships, as well as the emotional impact of their experiences.

**Adapting to new responsibilities and roles.** Participants reported a range of new responsibilities that they adopted at the time or in the aftermath of the trauma, primarily in order to care for their partner due to the PTSD and, in some cases, physical difficulties arising from the trauma. Although not all participants labelled themselves as a "carer", they all described new caring responsibilities and most said that they did the majority of household chores nowadays. Several participants reported taking responsibility for trying to manage their partners' PTSD symptoms. For some, such new responsibilities were so marked that they characterised shifts in roles within the relationship. Three participants likened this to becoming a mother caring for a child:

> I just wanna pick him, just wanna like wrap my arms around him and soothe him and say "Oh, shhh, you'll be okay", you know, like you would with a child . . .That's what the bulk of my life is like with him now. It's just wanting to take care of him, wanting to shield him and wanting to protect him from the big, bad, nasty outside world that makes him jump. (Laura)

Participants who described a shift in roles depicted it as being largely unwanted, with Kerry and Laura expressing particularly strong aversion for their new caring roles and worry that they could maintain them for a limited time only. The following extract demonstrates how Kerry adapted her parenting style in order to compensate for her husband's loss of skills. By adopting more traditional fatherly responsibilities, she was left feeling that she had surrendered motherhood:

> I was the one who would march in and say "Right! What's this? Do your homework! Do that!" . . .I felt I'd given my right up to motherhood really. It felt I'd become a father.

There was a sense that participants bore the brunt of the PTSD and that this was a stressful, exhausting and arduous position that they were compelled to assume. It was particularly anger and irritability that participants tolerated, which increased their own stress levels. For participants with children, the whole household bore the brunt of the PTSD. Several participants described being at the end of their tether at times and portrayed a sense that their coping resources were limited:

*Nicola's right you see, I haven't got any sympathy at the minute [laughs], not that she needs sympathy all the time, she doesn't. But I don't think Nicola likes to tell me when she's feeling particularly bad, because I go "Oh for god's sake" . . .Sometimes I'm just up to here [gestures]. (Sally)*

*Subordinated in the relationship*. A sense of feeling subordinated in the relationship emerged from the data, as though participants' positions had become less important as their partners' needs had increased. Narratives recounted how equality was lost and the relationship became one-way as the focus was placed wholly upon the partner. Sally described how she instinctively adapted when her partner returned from hospital, although she had not recognised this shift in the relationship until the interview:

*It's all about her . . .I hadn't thought that it had affected me, put it that way, but I think talking to you, probably you know it's affected me more than I thought . . .Every human adapts, don't they, to what they've got? And since she's come home from hospital, I think I've just adapted and I just, eurgh, slot where I feel I should be and do what I feel I should do.*

As part of this subordinated, undervalued position, some participants reported no longer being able to access emotional support from their partners. There was a sense that participants' emotional needs were neglected within the relationship, because their needs were considered less important:

*You think "Yeah okay I've listened to that long enough, I want to talk about me now, 'cause I've got a problem at work" [laughs]. And you don't get the chance to, because . . .his issues were bigger and so . . .tend to take precedence. (Eve)*

Subsequently, many participants perceived themselves as having lost personal qualities, such as their sense of humour. Laura described how the metaphorical loss of her partner had resulted in her being unable to talk about her feelings anymore and deemed this change so intense, she considered her personality had altered:

*The one person that I want to talk to is him and I can't. And he said to me the other day "I wish you'd talk to me Laura, I wish you'd talk to me more". And I said to him "How can I?". . .It's changed me; it's changed my personality, because I've always been somebody who will talk.*

*The emotional journey*. Participants described a vast array of emotions that they experienced at the time of the trauma and beyond, including disbelief, horror, distress, fear, worry, anger, self-blame, disappointment, resentment, pity and compassion. Guilt was a prevalent emotion that all participants reported or suggested experiencing. The following extract demonstrates how Laura made sense of her guilt, relating it to feeling selfish for experiencing her own suffering when she thought the focus should be on her partner:

*I feel that there's nothing mentally wrong with me, so I should be able to cope and I feel guilty when I'm having a bad day and for feeling bad 'cause I think "Well whatever he's going through is a million times worse than anything I'm feeling" . . .I feel selfish for wanting it to go away for my sake rather than for his sake.*

For a couple of participants whose partners suffered significant physical injuries, there was a strong sense of the trauma being traumatic for them too:

*Seeing him in the hospital, it was just, I mean he was grotesque . . .He just looked horrific. (Eve)*

There were many instances whereby participants described emotions that seemed to mirror their partners' emotions and PTSD symptoms. Some participants reported emotional and physiological responses (e.g. anxiety, sleeplessness) as a result of their own stress and shock. Others reported emotions such as irritation, low mood and happiness arising in response to their partner experiencing the same emotion. A couple of participants reported experiencing avoidance and hypervigilance in relation to their partners' PTSD triggers, because they were attempting to control the PTSD:

*I'm hypervigilant, I'm looking out for blooming dogs and I'm doing this and I'm trying to take the stress out of his life. (Kerry)*

*Finding ways to survive the experience.* Participants described a range of coping strategies employed in an attempt to survive their experiences. Many talked about how work and a life outside the house, away from their partner and the PTSD, were important in managing their own emotional response:

*I can remember thinking "Thank God I'm here", because you just get absorbed in work. . .Getting out of the house was my saving [laughs] grace, you know, from a selfish point of view. (Eve)*

Engaging in enjoyable activities and distraction techniques were utilised by many participants:

*I can get through it by doing other things, you know, taking the car and go for a ride. (Roy)*

A couple of participants resorted to unhelpful or unhealthy methods of coping, for example drinking alcohol excessively. At times, coping involved avoiding or disregarding their thoughts and feelings. At others, it involved acceptance of the situation. Despite using numerous methods to try and survive their experiences, the situation sometimes seemed too intense and participants' coping methods inadequate. Some acknowledged that they might appear to be coping outwardly, but were not coping internally. In this way, the image of coping was like a facade to the outside world, which was unaware of the trauma's true impact.

**3. Accessing and experiencing outside resources.** This theme is about participants' perceptions and experiences of support available to them, which ranged from being very helpful to very unhelpful. Support from outside the relationship was particularly pertinent, as many participants considered they had lost their partners' support within it. Accessing resources was complicated by factors such as stigma, ambivalence and desire for privacy.

*Support from others.* Experiences of emotional and practical support varied hugely both within and between participants. Many described how talking to friends, family and work colleagues was helpful. Eve described how the process of venting, receiving sympathy and obtaining new perspectives from friends and family was helpful for her, particularly because inside the home the focus was wholly on her husband:

*Being able to get it off your chest and say to somebody, "This is happening" . . .Also to get somebody else's perspective, you know, 'cause sometimes they would say: "Yes, but X, Y and Z, he has done this, this and that" and maybe that makes you step back and you think "Yeah, I could be a bit more understanding". Or other times, just to be sympathetic to me, because I suppose it was all very much about him at home.*

The relationship with support was complex, with most participants expressing how they needed or valued support, but certain factors prevented some from accessing it. For example, Sally's partner seemed ashamed of the stigma associated with PTSD and did not want their experiences discussed with others. Despite varied experiences of outside support, participants often reported feeling like they were coping alone.

Kerry found practical support from friends helpful at the time of the trauma (e.g. taking care of the children), but felt more ambivalent about accessing emotional support. She disliked discussing her problems and although people knew about her partner's difficulties, unlike Sally, they did not understand the full extent of them. She talked about the process of having and losing support:

*We were regular church goers . . .After Harry's injury, there was [laughs], they were almost offended that they couldn't heal him [laughs], so that aspect of our life went as well. . .and with that went a certain section of your friends or so-called friends [laughs] . . .It's a dangerous mix, you know, people thinking they can heal people and make people feel better and whatever and then writing them off. It just added to the hurt really.*

*Professional support.* Participants' opinions of professional support varied greatly with some aspects being much appreciated and others perceived as unhelpful or inappropriate. These included mental health and other services, for example physical health services, the Ministry of Defence and the police service. Participants identified ways in which professional support could have made the journey easier, particularly through increased awareness, greater communication and identifying and meeting their needs more effectively.

The need for increased awareness about PTSD in services that treat people with physical and psychological trauma was expressed, and for services to communicate how trauma can affect people. Kerry described how an information leaflet might help, akin to those used in physical healthcare:

*A bit like a head injury sheet you get, you know: "If you hear noises in your ears, get blood from your nose, go back". You almost need, you know: "You've suffered a traumatic event, if this, this and this happens, please refer back to your GP". I think would help so many people.*

Participants described ways that services could improve support offered to their partners, which in turn would impact upon themselves. These included a need for earlier intervention, greater attempts at engagement and further support out-of-hours. Several participants described stigma as creating ambivalence in their partners, preventing them from seeking

support, and articulated the need to dispel the shame and stigma associated with mental health difficulties:

> *If he hadn't have felt the stigma with [PTSD], he may well have made the connection sooner and said to them, "Well it might have been, it's been since I got back from [the warzone], maybe it's something to do with that". But because he was so in denial ...He didn't want it to be that, that was the last thing on earth he'd wanted it to be. (Laura)*

Participants reported having little or no involvement in their partners' mental health care. A wish for services to communicate with them more throughout their partners' treatment was expressed by some.

No participants reported having been offered any direct support themselves from NHS services. Some displayed ambivalence about professional support, considering its value as limited, or thinking that the practicalities of attending would increase their stress levels. Others emphasised the importance of professionals offering support and some thought that group support would be helpful, for example through normalising and validating their emotional responses:

> *It would be helpful just to talk to other people that are in my situation ...see how they cope, be reassured that my feelings of guilt and feelings of being selfish aren't unique and that I'm not selfish and I shouldn't feel guilty. (Laura)*

## Discussion

This study explored how partners of individuals in mental health services experience their loved ones' trauma and subsequent PTSD. The results showed that partners experience these as an ongoing journey of loss and gain, characterised by striving and struggling to make sense of the trauma and its ensuing consequences, whilst grappling with the identities of themselves, their partners and relationships shifting over time. Participants navigated their ongoing journeys in the context of external resources and support from friends, family and professionals.

Our results may inform theories around transitions and trauma. In the process of making sense of their experiences, participants emphasised numerous losses, including the loss of their partner and relationship, despite still being in the relationship. At times, these losses corresponded with a dismantling of expectations surrounding their life course. Parkes' psychosocial transitions theory suggests that transitions affect our set of assumptions about the world, known as our *assumptive world*, including everything we know or think we know, interpretations of the past and expectations of the future [29]. Similarly, Janoff-Bulman's Shattered Assumptions Model stipulates that trauma shatters fundamental assumptions about the self, others and the world [30]. Our results correspond with these theories surrounding transitions [29] and trauma [30], but extend both by suggesting that it is not only the individual's assumptive world that can be shattered through trauma, and need abandoning and restructuring in order to adjust successfully, but that of the partner too.

Perceived positive consequences of PTSD emerged from the data, which can be understood within the concept of posttraumatic growth [31]. Within this model, people can experience an increased appreciation for life and more meaningful interpersonal relationships. The model focuses on individual growth following trauma, but explains how supportive others can aid cognitive processing and assist in growth. Subsequent research has observed vicarious posttraumatic growth in partners of people with serious physical health problems [32] and professionals working with trauma survivors, including clinicians [33] and interpreters [34]. The

posttraumatic growth model has been expanded to the family system level, demonstrating how the family unit may experience growth following trauma [35]. Nevertheless, there remains limited understanding of the mechanisms by which PTSD creates vicarious opportunities for growth [36], although an interpersonal process including responsiveness and perceived responsiveness of one's partner may be one way, as indicated in a study investigating posttraumatic growth in couples whose homes had been severely damaged by flooding [37]. Further research into vicarious posttraumatic growth is warranted, including when and how partners' growth might influence one another in a couple system.

Consistent with previous research in partners of people with PTSD [38], participants reported experiencing a vast array of emotions. These emotional and psychological consequences of the trauma were akin to the impact described in the literature on carer burden in psychosis [39, 40] and dementia [41].

Participants described many instances whereby their emotions seemed to mirror their partners' emotions and PTSD symptoms. This occurred in three ways: emotional and physiological responses to stress which paralleled, but did not appear to be caused by, partners' PTSD; emotions arising in response to partners' specific emotions; avoidance and hypervigilance arising from attempts to manage the PTSD. These mirroring processes indicate the complexities underpinning participants' emotional experiences and might illuminate existing literature on secondary and vicarious traumatisation. Results from the present study may provide some empirical support for the mechanisms of systemic traumatic stress proposed within Nelson Goff and Smith's Couple Adaptation to Traumatic Stress model [42], including chronic stress, empathy, identification and internalisation. However, participants' reports of PTSD-like symptoms due to conscious (or perhaps unconscious) efforts to manage their partners' PTSD do not appear to reflect any of the proposed mechanisms and thus may further inform the model.

## Implications for services

The findings of this study indicate that participants experienced a range of losses and emotions. Participants highlighted their desire to be more informed and involved in their partners' treatment. Through their inclusion in the assessment process, partners may be signposted to relevant statutory and non-statutory services to better support their own needs. The development of interventions for partners specifically could be helpful. A one-to-one or group space to receive psychoeducation, make sense of emotional responses, and have them normalised, might be helpful for some partners in facilitating adjustment and promoting wellbeing. In a systematic review of interventions for supporting partners of military veterans with PTSD, psychoeducation related to communication, problem solving, and emotion regulation was suggested to be particularly important [43]. Alternatively, service providers could support the creation of partner-led safe spaces or networks for partners to discuss their experiences with other partners. Indeed, research in Australia suggests that support groups for female partners of Vietnam veterans can be very helpful [44].

Similarly, couples therapy has been suggested as a helpful intervention for couples affected by PTSD, including Emotionally Focused Therapy for couples [45] and Cognitive Behavioural Conjoint Therapy [46]. In a study whereby six couples (where one partner in each couple had PTSD) received Cognitive Behavioural Conjoint Therapy (CBCT) with methylenedioxy-methamphetamine (MDMA), posttraumatic growth and social intimacy was improved in both partners [47].

From a health economic perspective, whilst the suggested service developments would require additional resources and clinician time, managing psychological distress in partners

might have a positive financial impact on the health service in the long-term, preventing partners from entering mental health services themselves at a later stage due to their own mental ill health.

## Strengths and limitations

This study provides novel, rich and in-depth insight into the experiences and views of partners of people who have experienced trauma and been diagnosed with PTSD. We included a relatively diverse group of partners, including both sexes and one couple from a same-sex relationship. We have adhered to rigorous and transparent procedures for the conduct of IPA and good quality qualitative research.

Although the researchers took several steps to ensure the credibility of the study, in line with guidelines suggested by Yardley [24], it is generally considered impossible for researchers to suspend their own assumptions and biases [48]. The first author's preunderstanding would inevitably have influenced lines of questioning at interview and construction of meanings at analysis. This influence was tempered by other members of the research team reading the transcripts independently and discussing the developing analytical framework together.

The findings of this study must be considered within the context of its limitations. This study took place with partners of services users from one specialist Trauma Service in the UK and all participants identified as white British, so the data is limited to this particular cultural group. Further research is necessary to extend this study's findings and explore the experiences of a more demographically diverse group of partners. Such research may inform the development of services that can provide culturally-sensitive support to partners and families of people with PTSD.

As most previous research with partners has tended to be conducted with female partners of individuals in male-dominated professions (e.g. military samples), male partners are a highly under-researched group, as are those in same-sex relationships. More research is needed to explore further the experiences and perspectives of people from these groups. Finally, the study intentionally excluded the experiences of people who were no longer in their relationship. By not giving voice to these people, the processes by which trauma and PTSD might influence relationships ending may have escaped the scope of this research.

## Conclusions

The current study adds to a limited body of research about the systemic effects of trauma, helping to enrich understanding of the relevant issues for families, inform service provision and guide further research. An in-depth analysis of partners' experiences describes how partners made sense of their loved ones' trauma and its consequences within the context of their relationship. The voices of partners are often overlooked in research and clinical services, particularly those with partners outside of high-risk occupations. The present study gives voice to partners' experiences and sheds new light on the impact of being a partner of a person who has experienced trauma and been diagnosed with PTSD. Whilst there was rich diversity amongst participants' stories, one overarching theme emerged that encapsulated the data: participants experienced their loved ones' trauma and PTSD as an ongoing journey of loss and gain. The need for services to provide greater information and support for partners was highlighted. Further research is needed into systemic traumatic stress and growth, how these influence one another in a couple system, and the ways in which services can provide effective, meaningful and culturally-sensitive support to families of people with PTSD. In the context of limited resources, further consideration by services to the systemic effects of trauma and how to support the system around the individual is warranted.

## Supporting information

**S1 File. Interview schedule including demographic and contextual questions.** This document outlines the demographic and brief contextual questions participants were asked regarding their partners' trauma and PTSD and their relationships, and the interview schedule which was used to guide the semi-structured interviews.
(DOCX)

## Acknowledgments

The authors thank all those who assisted with this study, in particular the individuals who participated in interviews and their partners. We also wish to explicitly thank the Service User and Carer panel at the University of Surrey who provided feedback regarding ethical considerations and study materials.

## Author Contributions

**Conceptualization:** Rosie Powling, Dora Brown, Jo Billings.

**Data curation:** Rosie Powling.

**Formal analysis:** Rosie Powling.

**Investigation:** Rosie Powling.

**Methodology:** Rosie Powling, Dora Brown, Jo Billings.

**Project administration:** Rosie Powling, Dora Brown, Jo Billings.

**Resources:** Rosie Powling, Dora Brown, Jo Billings.

**Supervision:** Dora Brown, Jo Billings.

**Validation:** Rosie Powling, Dora Brown, Jo Billings.

**Visualization:** Rosie Powling, Sahra Tekin, Jo Billings.

**Writing – original draft:** Rosie Powling.

**Writing – review & editing:** Rosie Powling, Dora Brown, Sahra Tekin, Jo Billings.

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
