## [Decision Letter · Decision Letter 0]

11 Apr 2023

PONE-D-23-01129Partners' experiences of trauma and PTSD: An ongoing journey of loss and gainPLOS ONE

Dear Dr. Powling,

Thank you for submitting your manuscript to PLOS ONE. After careful consideration, we feel that it has merit but does not fully meet PLOS ONE’s publication criteria as it currently stands. Therefore, we invite you to submit a revised version of the manuscript that addresses the points raised during the review process.

We look forward to receiving your revised manuscript.

Kind regards,

Michal Ptaszynski, PhD

Academic Editor

PLOS ONE

Journal Requirements:

https://journals.plos.org/plosone/s/file?id=ba62/PLOSOne_formatting_sample_title_authors_affiliations.pdf"

Reviewers' comments:

Reviewer's Responses to Questions

**Comments to the Author**

1. Is the manuscript technically sound, and do the data support the conclusions?

Reviewer #1: Partly

Reviewer #2: Yes

2. Has the statistical analysis been performed appropriately and rigorously? 

Reviewer #1: N/A

Reviewer #2: Yes

3. Have the authors made all data underlying the findings in their manuscript fully available?

Reviewer #1: Yes

Reviewer #2: No

4. Is the manuscript presented in an intelligible fashion and written in standard English?

Reviewer #1: No

Reviewer #2: Yes

5. Review Comments to the Author

Reviewer #1: This research is important in clarifying that not only the individual person with trauma and PTSD suffer from the consequences. Also their partners are suffering and need support. It is obvious, that the researcher has gained in-depth insight into the experiences of the partners. The results are very detailed and the connection between the themes developed and the participants' voices is clear. However, the manuscript has flaws and a major revision is needed.

Abstract: You state that you aim is to explore "such experiences". Please be specific. Overall, the abstract will need revision after the main manuscript has been revised.

Title: Regarding the first part of the title "Partners' experiences of trauma and PTSD": It suggests that you are studying partners' experiences of trauma and PTSD. PTSD is a disorder, the diagnosis given to a person. However, your research is concerned with people. I suggest that you are studying the experiences partners have of being in a relationship with people who have suffered trauma and PTSD? As you even suggest in the first part of the discussion (l. 514): "This study explored how partners of individuals in mental health services experience their partners’ trauma and subsequent PTSD". Please revise accordingly.

The second part of the title is intriguing.

Introduction: The review of the literature is satisfactory. However, the narrative is a bit confusing to read.

Particularly the section l. 95- 102 is not coherent, confusing to read, and does not contribute to stating the knowledge gap. Please revise to clarify. In l. 102-105 is not supported by evidence; a reference is lacking.

The research question and aim are unclear. In revising according to the following comments, it may be useful for you to apply the PICo mnemonic in structuring your research question and aim:

• You state that partners' experiences of trauma and PTSD is the focus of the research. In the same way as in the title this wording suggests that you are studying partners' experiences of symptoms and diagnosis, rather than studying experiences of being in a relationship with a person with PTSD. Reading your introduction, it seems that you are interested in partners' experiences of living together with a person with trauma and PTSD; or partners' experiences of the relationship?

• Further, introduction is missing to the key subjects; trauma and PTSD. Are they the same? Or how does trauma develop into PTSD? To be part of your study should the person who experiences a trauma be diagnosed with PTSD?

• You state that there is a significant gap in the literature in exploring the perspectives of partners from outside of high-risk occupational settings. However, your research questions do not reflect this gap. Further, your focus on this particular group should be reflected in the "Participants" section.

• In l. 141 you introduce the term "service user" for the first time. Who are they? In what context are they service users? This term should be clarified in the introduction. I wonder whether rewriting "service users" would account for the Context part of the PICo mnemonic in describing the research question?

• Concerning the language: l. 107 you write: "… with little qualitative research hearing partners’ experiences and views in depth." Setting out for the formulation of the aim, "hearing" seems a bit odd. In qualitative research the aim is to explore, understand, investigate. In describing the aim, you use "participants" (l. 112). Please, reformulate so the aim reflects the population.

Overall; the use of parenthesis is confusing: do they signal that their contents are less important? In which case it could be left out of the text. Particularly in two places the problems / symptoms of the person with PTSD is put in parenthesis. These experiences are very important to be able understand the situation of the partners. A short introduction to the person who has been diagnosed with PTSD, is needed to fully understand the challenges meeting the partners.

In l. 48-49 you write that perceived social support is protective factor in the development of posttraumatic stress disorder. This is a bit confusing; it indicates that social support is particularly important after having had a trauma to avoid developing PTSD. While it seems that your study is focusing on being in a relationship with a person who has developed PTSD?

l. 50-51: "…the impact on those providing such support…" –The impact of what? What is it that impacts on the person providing support? Please clarify.

l. 52-53: "…a limited number of studies have explored how trauma affects partners and relationships". What trauma are you writing about? The partners of whom? Do you mean being in a relationship with a person who has experienced a trauma – or who has developed PTSD? Please, clarify.

L. 53-58: "Research has tended to use quantitative methodology…": the research you are referring to actually used quantitative methodology. Writing about tendencies is a bit odd, when you want to introduce the knowledge we have gained from quantitative research.

l. 79: "This research reports similar deleterious effects of being the partner of …". Similar to what?

L. 80: "Someone": I suggest you write "a person".

In terms of content, the introduction is long with the use of redundant words. I suggest a revision focusing up on tightening up the language; i.e. less referring and more directly stating the results.

Three examples with suggestions:

• L 52: "To date, a limited number of studies have explored how trauma affects partners and relationships. Research has tended to use quantitative methodology and focus on 54 specific traumas and populations at high risk of exposure to trauma (most commonly 55 combat-related trauma in male military personnel and veterans)". A more focused version e.g. could be: "Few studies have explored how trauma affects partners and relationships. Quantitative research has focused on specific traumas and populations at high risk of exposure to trauma, mainly combat-related trauma in male military personnel and veterans.

• l. 62: Male soldiers’ symptoms (particularly sleep problems, dissociation and sexual problems) have been demonstrated to predict lower relationship satisfaction for both soldiers and their female partners". A more focused version e.g. could be: "For male soldiers’ and their partners' symptoms such as sleep problems, dissociation and sexual problems can predict lower relationship satisfaction."

• l 77: "An even smaller body of literature has qualitatively explored partners’ experiences, with what little research that has been conducted again mostly being with spouses of workers in high-risk occupational groups". A more focused version e.g. could be: "Few qualitatively studies explored partners’ experiences. Most studies concerned spouses of workers in high-risk occupational groups".

In the section l. 79- 85 you repeatedly describe qualitative studies reporting on effects and outcomes. I suggest revising the text so that it reflects the reference to qualitative research. Examples:

• l. 80: "… similar deleterious effect of being a partner...". I suggest you describe how being a partner was deleterious (l 80).

• l. 81: "In one qualitative study exploring heterogeneous traumas, a history of trauma in one or both partners had both positive and detrimental effects on interactional patterns in the relationship, including increased and decreased cohesion, communication and understanding". To highlight experiences / perceptions this you "A history of trauma in a study exploring heterogeneous traumas was reported to both increase and decrease experiences of cohesion, communication, and understanding in partners' interaction"

Methods

Participants

l. 127: use the abbreviation PTSD

l. 127-128: What is a "type 1 trauma"? A definition is lacking.

Procedure

l. 135: "A Service User and Carer panel at the University of Surrey 136 was consulted regarding ethical considerations". These considerations are important. Please, elaborate. For example, I expect that the lengthy individual interviews have called for certain considerations?

l. 141: How and by whom were potential participants who had not been in direct contact with the Trauma Service approached (after informed consent from service users were obtained)?

l. 146: To make visible the researchers (first authors) pre-understandings, please elaborate on relevant characteristic; i.e. qualifications, experiences.

l. 149: You use terms such as "pre-determined" and "open questions". Are these terms referring to IPA? If not, I suggest you use well-established concepts such as "semi-structured interview guide" and "open-ended.

l 153-156: rather than referring the guidelines suggested by Yardley, briefly describe measures to comply with relevant parts of these guidelines

Results

To provide readers with an overview, I suggest to display characteristics of the participants in a table and briefly present characteristics in text.

Discussion

In places you emphasize that experiences related to i.e. "some participants" or "a couple of participants". This should be rephrased to discuss the topics in more general terms.

l. 524-527: To better understand the needs of the partners it would be helpful if you briefly elaborated on the concept "assumptive world".

l. 544 – 547: I suggest to discuss and highlight part that are particular interesting in relation to professional practice.

Implications for services

l. 585: The chain of argument to conclude that "…the issue of considering partners is not simply a systemic issue, but a moral, political and financial one" is insufficient. Regarding the financial aspect, I suggest to provide some background in the introduction.

Strengths and limitations

The researcher's pre-understanding is of great importance for the credibility of data and should be addressed.

Conclusions

l. 613-615: This part of the conclusion is not supported by the results.

Reviewer #2: This study examined how partners of individuals affected by PTSD experience their partners’ trauma and subsequent mental health issues.

This is a study focusing on (mental health) an area of oppression for many because mental health issues are still a bit hard for many to discuss openly as a result of the stigma associated with it. It was good to have a couple in same sex relationship participate in this study.

I have suggested a few minor issues for the authors to address.

Implication for services:

all participants of the study identified as white so how do you then call for The ‘development of culturally-sensitive intervention’? This does not seem to have a very direct connection here.

Can your implications further target religious or faith-based approaches also?

since participants talked about the desire to have a group to discuss their partners’ PTSD, how about service providers introducing (support the creation of)safe spaces or networks to discuss these among partners?

6. PLOS authors have the option to publish the peer review history of their article (what does this mean?). If published, this will include your full peer review and any attached files.

Reviewer #1: No

Reviewer #2: No

---

## [Author Response · Author response to Decision Letter 0]

30 Jun 2023

Dear Michal Ptaszynski and Reviewers, 

Re: PONE-D-23-01129

Thank you very much for your comprehensive feedback on our manuscript, now entitled "Partners’ experiences of their loved ones' trauma and PTSD: An ongoing journey of loss and gain". We are pleased to provide a point-by-point response to your comments in our attached letter 'Response to Reviewers'. 

We would like to sincerely apologise for the delay in resubmitting the manuscript, after it was quickly returned to us in order to remove a couple of comments that had inadvertently appeared in the PDF. When the PDF was rebuilt, it did not appear to include the manuscript with track chances. Despite many attempts to rectify this, and several communications with the PLOS ONE office, we have unfortunately been unsuccessful at gaining a response as to how to rectify this issue or whether to approve the submission. Given the length of time that has passed, we have decided to approve the resubmission as it is, with the manuscript with track changes uploaded but not appearing in the PDF. We hope this is acceptable and trust that we will hear back if not, including how to rectify the issue. 

We hope the amendments address the points raised and we look forward to hearing further from you. Thank you very much again. 

Yours sincerely, 

Dr. Rosie Powling

Clinical Psychologist & BABCP Accredited Cognitive Behavioural Therapist

Clinical Education Development And Research, University of Exeter

---

## [Decision Letter · Decision Letter 1]

18 Jul 2023

PONE-D-23-01129R1Partners' experiences of their loved ones' trauma and PTSD: An ongoing journey of loss and gainPLOS ONE

Dear Dr. Powling,

Thank you for submitting your manuscript to PLOS ONE. After careful consideration, we feel that it has merit but does not fully meet PLOS ONE’s publication criteria as it currently stands. Therefore, we invite you to submit a revised version of the manuscript that addresses the points raised during the review process.

We look forward to receiving your revised manuscript.

Kind regards,

Michal Ptaszynski, PhD

Academic Editor

PLOS ONE

Journal Requirements:

Reviewers' comments:

Reviewer's Responses to Questions

**Comments to the Author**

1. If the authors have adequately addressed your comments raised in a previous round of review and you feel that this manuscript is now acceptable for publication, you may indicate that here to bypass the “Comments to the Author” section, enter your conflict of interest statement in the “Confidential to Editor” section, and submit your "Accept" recommendation.

Reviewer #1: (No Response)

2. Is the manuscript technically sound, and do the data support the conclusions?

Reviewer #1: Yes

3. Has the statistical analysis been performed appropriately and rigorously? 

Reviewer #1: N/A

4. Have the authors made all data underlying the findings in their manuscript fully available?

Reviewer #1: Yes

5. Is the manuscript presented in an intelligible fashion and written in standard English?

Reviewer #1: Yes

6. Review Comments to the Author

Reviewer #1: The authors have done a great job in improving the manuscript. However, some work still needs to be done.

Abstract: The abstract has become clear and easy to read. One thing should be amended: l.32 is part of the methods. Please, move to the Methods section of the abstract.

Introduction: The section has improved; however, the section is lengthy and the narrative still is somehow confusing to read. While the study is qualitative focusing on how better to understand people´s experience, the language in places should be rephrased to give a sense of coherence throughout the manuscript. Particularly on page 3 l. 61- 71, the use of nomenclature from quantitative research is disturbing, i.e. "correlation", and "effect size". I suggest a revision. In particular, in l. 69-71 the point is hard to see. I suggest you revise to highlight the focus on people with PTSD and partners rather than effect size.

An example of revising l. 61- 64 could read: "Further in veterans with more severe PTSD symptoms, their partners have increased distress, lower levels of functioning and poorer mental health".

In places, too much details makes the reading un-interesting without adding relevant knowledge. This pertains to highlighting different sub-groups of high-risk occupational groups; or the highlighting of the amount of research. One example is l. 72-75, that could be shortened to "Partners of fire-fighters experience similar symptoms such as major depression, panic disorder, and generalised anxiety disorder [10] and more recently, 72.5% of spouses of 76 doctors reported psychological distress during the COVID-19 pandemic [11]". Another example is l. 99-100: "…recruited 100 from a university-based counselling centre…": I do not see the relevance of this information; and I suggest you revise l. 99-103 to make your point clear.

Further, unnecessarily referring to existing research disturbs the reading; in many instances the reference is sufficient to show the evidence. Here are some examples:

• l. 52: "Meta-analyses have established that" - can be removed.

• lines 60 & 64 & 68 & 86: "have/has been shown to". I suggest you remove and where necessary rephrase.

• l 62: "Research has established": Remove and rephrase accordingly

• l. 94: "in one such qualitative study" – can be deleted.

Finally, unnecessary small words are used throughout the manuscript, making it longer than necessary and in places more confusing to read, for example:

• l. 55: "such": you indicate, that one specific earlier defined kind of social support is referred to. suggest you delete.

• L.56: "relatively": in relation to what? suggest you remove

• "Also": is used comprehensively throughout the manuscript, in places it has no meaning (i.e. l. 37, l. 82 and l. 458) and in places it is unclear what it is referring to (i.e. l. 79) Please go through critically.

• l. 565 "aforementioned" – suggest you delete

Other comments:

• L. 93: I do not understand that the experiences are "indicated"? The experiences referred to are, I guess, results from the studies?

• l. 106-108: It seems that some argument is missing in this sentence; Development of knowledge and understanding cannot in itself "increase partner's social connectedness …".

Methods

The section is in some places still referential, without contributing insight into the actual study, particularly in the section on credibility.

l 152: "was necessary" and l. 153: was required. Please, reformulate to describe what you actually did (i.e. consent was obtained)

l. 163-166: "A semi-structured interview schedule …. explored experiences of". An interview guide cannot explore; it is a means used to explore something. Please amend

L. 166: The sentence is repetitive, please delete

l. 173-175: can be deleted

l. 177: "It is argued that": Please delete. You actually supported commitment and rigor.

l 187-189: can be deleted

l. 189-191: If you want to keep this, it should be part of the discussion or conclusion.

Strength and limitation:

l 636-650: I do find this section to be too long and unnecessarily detailed and in places even referential (i.e. l. 637-640). With the detailed description in the methods section it is sufficient to write more concisely, for example l 640-642 could be abbreviated to "The first author's preunderstanding…". Please, revise.

Conclusion:

This sentence does not make sense: "… one overarching theme emerged that encapsulated the data, that is participants experienced trauma and PTSD as an ongoing journey of loss and gain". Please amend.

7. PLOS authors have the option to publish the peer review history of their article (what does this mean?). If published, this will include your full peer review and any attached files.

Reviewer #1: No

---

## [Author Response · Author response to Decision Letter 1]

30 Aug 2023

Dear Dr. Ptaszynski, 

Thank you to you and your reviewers for your feedback on our amended manuscript Partners’ experiences of their loved ones’ trauma and PTSD: An ongoing journey of loss and gain (PONE-D-23-01129R1). We are pleased to provide a point-by-point response to the comments in table-format in the ‘Response to Reviewers’ letter attached to this submission. Thank you again for considering our manuscript and providing such comprehensive feedback. We hope the amendments address the points raised and we look forward to hearing further from you.

Yours sincerely, 

Dr. Rosie Powling

Clinical Psychologist & BABCP Accredited Cognitive Behavioural Therapist

Clinical Education Development And Research, University of Exeter

---

## [Decision Letter · Decision Letter 2]

18 Sep 2023

Partners' experiences of their loved ones' trauma and PTSD: An ongoing journey of loss and gain

PONE-D-23-01129R2

Dear Dr. Powling,

We’re pleased to inform you that your manuscript has been judged scientifically suitable for publication and will be formally accepted for publication once it meets all outstanding technical requirements.

Kind regards,

Michal Ptaszynski, PhD

Academic Editor

PLOS ONE

Additional Editor Comments (optional):

Reviewers' comments:

Reviewer's Responses to Questions

**Comments to the Author**

1. If the authors have adequately addressed your comments raised in a previous round of review and you feel that this manuscript is now acceptable for publication, you may indicate that here to bypass the “Comments to the Author” section, enter your conflict of interest statement in the “Confidential to Editor” section, and submit your "Accept" recommendation.

Reviewer #1: All comments have been addressed

2. Is the manuscript technically sound, and do the data support the conclusions?

Reviewer #1: Yes

3. Has the statistical analysis been performed appropriately and rigorously? 

Reviewer #1: N/A

4. Have the authors made all data underlying the findings in their manuscript fully available?

Reviewer #1: Yes

5. Is the manuscript presented in an intelligible fashion and written in standard English?

Reviewer #1: Yes

6. Review Comments to the Author

Reviewer #1: The authors have done a great job in improving the manuscript, rendering it coherent and interesting to read. The methodological strengths of the study are now clearly visible.

7. PLOS authors have the option to publish the peer review history of their article (what does this mean?). If published, this will include your full peer review and any attached files.

Reviewer #1: No

---

## [Editor Report · Acceptance letter]

18 Jan 2024

PONE-D-23-01129R2 

PLOS ONE

Dear Dr. Powling, 

I'm pleased to inform you that your manuscript has been deemed suitable for publication in PLOS ONE. Congratulations! Your manuscript is now being handed over to our production team.

Kind regards, 

on behalf of

Dr. Michal Ptaszynski 

Academic Editor

PLOS ONE